

# Quasi-localized vibrational modes, boson peak and sound attenuation in model mass-spring networks

**Shivam Mahajan[1] and Massimo P. Ciamarra[1,2,3⋆]**

**1** Division of Physics and Applied Physics, School of Physical and Mathematical Sciences,
Nanyang Technological University, Singapore
**2** CNRS@CREATE LTD, 1 Create Way, #08-01 CREATE Tower, Singapore 138602
**3** CNR–SPIN, Dipartimento di Scienze Fisiche, Università di Napoli Federico II,
I-80126, Napoli, Italy

⋆ massimo@ntu.edu.sg

## Abstract

We introduce an algorithm that constructs disordered mass-spring networks whose elastic properties mimic that of glasses by tuning the fluctuations of the local elastic properties, keeping fixed connectivity and controlling the prestress. In two dimensions, the algorithm reproduces the dependence of glasses' vibrational properties, such as quasilocalised vibrational modes and Boson peak, on the degree of stability. The sound attenuation displays Rayleigh scattering and disorder-broadening regimes at different frequencies, and the attenuation rate decreases with increased stability. Our results establish a strong connection between the vibrational features of disordered solids and the fluctuations of the local elastic properties and provide a new approach to investigating glasses' vibrational anomalies.



# 1   Introduction

The density of vibrational states (vDOS) controls solids' specific heat and transport properties [1,2]. The vDOS of amorphous solids differs qualitatively from that of crystals. In crystals, the low frequency vibrational excitations are plane waves (phonons) distributed in frequency according to Debye's law, $D_p(\omega) \propto \omega^{d-1}$ in $d$ spatial dimensions. On the contrary, amorphous materials have an excess of low-frequency modes over Debye's prediction that induces a peak in the reduced density of states $D(\omega)/\omega^{d-1}$ at the Boson peak frequency, $\omega_{\mathrm{bp}}$, in the terahertz regime for molecular solids. Previous works have attributed the Boson peak to elastic disorder [3–5], localized harmonic/anharmonic vibrations [6–9], broadening of van Hove singularities [10,11] (but see [12]). In addition, in amorphous solids the low-frequency excitations comprises both phononic-like modes and additional quasi-localized vibrational modes (QLMs) that appears to be universally distributed in frequency as $D_{\mathrm{loc}}(\omega) = A_4 \omega^4$ [13–16], with $A_4$ decreasing as the stability of the material increases [17]. The low-frequency $\omega^4$ scaling of the density of soft-localized modes may originate from general considerations on the properties of localized excitations in disordered systems [18] (see, however [19]). The universality of the scaling law [14,15] and its dimensionality independence suggest that this scaling has a mean-field origin [20–22]. Finally, in amorphous materials the extended low-frequency modes are not phonons: even in the absence of temperature induced anharmonic effects [23], phonons of wave vector $\kappa$ attenuate with a rate $\Gamma(\kappa)$ exhibiting a crossover from a Rayleigh scattering [24] regime, $\Gamma \propto \kappa^{d+1}$, to a disordered-broadening regime, $\Gamma \propto \kappa^2$ [25–29], as $\kappa$ increases.

The squared vibrational eigenfrequencies $\omega^2$ are the eigenvalues of the matrix of the second derivatives of the energy with respect to the particle positions or Hessian matrix. As such, the vibrational anomalies of amorphous materials may possibly be rationalized within random matrices [30–35]. Previous works primarily focused on the eigenvalues of Wishart matrices, which are positively defined and hence may model stable systems. A mean field [36] random-matrix approach suggests that the Boson peak may originate from the reduction in coordination number driving the system toward isostaticity [31, 37] and from hierarchical energy landscape. These two scenarios are possibly relevant in colloidal hard-sphere-like glasses and highly connected molecular systems. The random matrix approach may also be used to investigate QLMs. In this case, the issue is determining the random matrix ensemble reproducing the $D_{\mathrm{loc}}(\omega)$ distribution characterizing amorphous solids or, equivalently, the correlations to be enforced on the matrix. Research in this direction [38] succeeded in reproducing a pseudo-gap, $D(\omega) \propto \omega^{\alpha}$, with an exponent $\alpha < 4$. In this research direction, the issue is integrating the two approaches to random matrices that reproduce at the same time Boson peak and quasi-localized modes, as well as their correlations.

Other approaches recovered the $\omega^4$ distribution by describing an amorphous material as an elastic continuum punctuated by defects, possibly anharmonic or interacting [21, 22, 39, 40]. Localized vibrations may thus cause all vibrational anomalies of glasses, considering that they may induce the Boson peak [6–9] and control sound attenuation in Rayleigh's theory [24]. Recent numerical results supported this scenario in three dimensions by demonstrating a relation between QLMs' frequency distribution and Boson peak frequency [17, 41], $A_4 \propto \omega_{\mathrm{bp}}^{-5}$.

In two-dimensions, vibrations with frequencies close to the Boson peak consist of phonons hybridized with QLMs [42].

In this manuscript, we investigate the physical origin of the vibrational anomalies of amorphous solids by creating mass-spring networks, or equivalently Hessian matrices, that reproduce them and their relationships. Rather than looking for the ensemble of random matrices exhibiting the anomalies of interest, we study how to vary an amorphous solid's mass-spring network to modulate them. Similar approaches have been introduced to induce a Boson peak in unstressed networks [43] or suppress $A_4$ by artificially reducing the prestress in stressed ones [44]. Here, we introduce an algorithm to generate systems with varying degrees of elastic disorder and prestress. We find that the boson peak frequency $A_4$ satisfies the relation $A_4 \propto \omega_{\rm bp}^{-5}$ and that $A_4$ decreases as the degree of elastic disorder is suppressed. Our networks also reproduce sound's attenuation crossover from a Rayleigh scattering to a disordered broadening regime observed in amorphous solids. In the Rayleigh scattering regime, the attenuation rate relates to the material properties as predicted by fluctuating elasticity theory [3, 4], consistently with our observation of no varying correlations in the elastic properties. Our results support a deep connection between the vibrational anomalies and glasses and clarify their relationship with the fluctuations of the local elastic properties.

The paper is organised as follows. We introduce our approach to construct mass-spring networks in Sec. 2. We illustrate how it influences geometrical and macroscopic elastic properties in Sec. 3, and the local elastic ones in Sec. 4. Sec. 5 demonstrates that the vibrational spectrum of our networks reproduces the vibrational anomalies of amorphous materials and their correlations [41]. Finally, Sec. 6 shows that sound attenuation in our networks crossovers from Rayleigh scattering regime $\Gamma \sim \omega^3$ to disorder-broadening regime $\Gamma \sim \omega^2$, as expected for two-dimensional solids. We summarise our results and discuss future research directions in the conclusions.

## 2 Numerical model and protocols

Our mass-spring network generating algorithm takes as input the disordered mass-spring network associated with the linear response regime of an amorphous solid, which generally has bond-depending elastic constants and rest lengths. We transform this original network by swapping the attributes of randomly selected bond pairs, i.e., by exchanging their spring constants and rest lengths, as schematically illustrated in Fig. 1. Henceforth, the algorithm does not vary the connectivity. We define the fraction of swapped bonds as $f = 2N_{\rm swap}/N_b$, where $N_{\rm swap}$ is the number of swap moves, $N_b$ the number of bonds in the network, and the factor 2 accounts for the fact that each swapping event involves two bonds. Hence, for $f = 0$ we retain our original network, while for $f = 1$ each bond has been swapped once on average. After the bond swapping, we minimize the energy of the new network bringing it into a mechanically stable configuration and study its vibrational properties.

We remark that, on increasing $f$, the bond randomization procedure destroys the correlations in the local elastic properties of the initial network more effectively. However, regardless of the $f$ value, the elastic properties of the final network might or might not exhibit correlations that build up during the final minimization procedure. The exact relation between $f$ and correlation in the elastic properties, if any, needs to be determined a posteriori.

The swapping procedure may change the network's prestress. To ascertain if the changes in the prestress correlate with changes in the vibrational properties, for selected $f$ values we also generate a set of networks at fixed prestress, which we tune by varying the density after the bond-swapping procedure.

We have implemented our bond-swapping procedure in two dimensions. To generate our

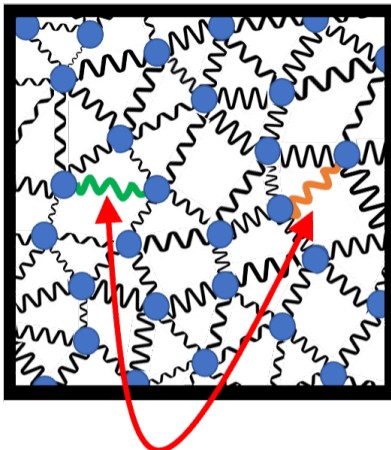

Figure 1: Schematic of the bond-swapping algorithm we use to tune the elastic properties of a disordered mass-spring network. In each bond swapping event, we randomly select two elastic springs and swap the values of their elastic constants and rest lengths. We obtain different network by varying the fraction of swapped bonds $f = 2N_{\text{swap}}/N_b$, where $N_{\text{swap}}$ is the number of swap moves and $N_b$ the number of bonds in the network, and then minimizing the energy of the resulting network.

initial network, we consider systems of particles interacting via the Weeks-Chandler Andersen (WCA) potential

$$U_{ij}(r_{ij}) = 4\epsilon \left[ \left( \frac{\sigma_{ij}}{r_{ij}} \right)^{12} - \left( \frac{\sigma_{ij}}{r_{ij}} \right)^{6} \right] + \epsilon , \qquad r_{ij} \leq 2^{1/6} r_{ij} , \qquad (1)$$

with $r_{ij}$ the distance between interacting particles, $\sigma_{ij} = (\sigma_i + \sigma_j)/2.0$ where $\sigma_i$ is a particle diameter drawn from a uniform random distribution in the range [0.8:1.2]. We equilibrate systems at fixed number density $\rho = 1.2$ at high temperature $T = 4\epsilon$, and then instantaneously quench them into amorphous solid configurations by minimizing the energy via the conjugate-gradient algorithm [45].

We generated initial mass-spring networks with $N = 1024$ to 360000 particles. Each network is fed to our algorithm to create different networks by swapping a fraction $f$ of the bonds. For each $N$ and $f$, we average our data over 200 independent initial networks unless otherwise specified.

## 3 $f$ dependence of mechanical and geometrical properties

We investigate the influence of the bond-swapping procedure on the geometrical and mechanical properties of the elastic network in Fig. 2. Panel a shows that bond swapping does not affect two-point correlations as the radial distribution function is de-facto $f$-independent. In panel b, we study the $f$ dependence of the distribution of the interparticle forces F. We obtain our initial $f = 0$ network by minimising the energy of a system of particles interacting via a repulsive potential. Consequently, for $f = 0$ all interparticle forces are positive, i.e., repulsive. The swapping protocol changes the interparticle forces' distribution by inducing tensile forces. These changes influence the network's mechanical properties by increasing the shear modulus and reducing the pressure, as in Fig. 2c. The reduction in pressure results from the emergence of tensile forces. To rationalise the origin of the shear modulus' change, we decompose $\mu_0 = \mu_{\text{a}} - \mu_{\text{n.a.}}$ in its affine and non-affine contribution. We find $\mu_0$ increases with $f$ as the

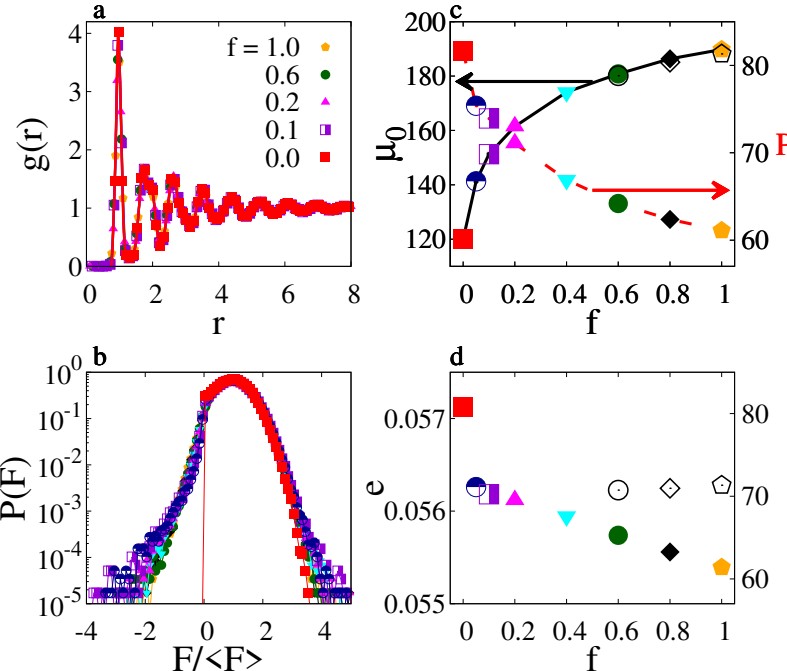

Figure 2: Panel (a) illustrates the radial distribution function of $N = 40000$ particle systems. Panel (b) shows the distribution of the magnitude of the spring forces $F$ normalized by their average value. Data are averaged over 200 configurations. Panel (c) illustrates the dependence of shear modulus $\mu_0$ on the swapping fraction $f$. Panel (d) shows the variation of pre-stress $e$ with $f$. Data (b)-(d) are for $N = 1024$. Full symbols refer to different $f$ values, e.g., as in (b). Open symbols for $f \geq 0.6$ represent systems compressed to impose a fixed pre-stress value.

affine contribution grows while the non-affine contribution decreases.

As the swapping fraction $f$ increase, the prestress [46,47] $e=(d-1)\langle(-U'(r_{ij})/r_{ij}U''(r_{ij})\rangle_{ij}$ monotonically decreases, as illustrated in Fig. 2d. The prestress change is small and comparable to that reported in previous works [44,48], and mostly occurs on moving from $f = 0$ to $f > 0$, i.e., as tensile forces appear in the system. To assess if this small change in prestress could sensibly affect the vibrational properties, at selected $f$ values we create additional networks by changing the density to impose a fixed prestress value. In Fig. 2d, we use open symbols to indicate the pre-stress value of these configurations.

## 4  Disorder parameter and fluctuations of elastic properties

Our investigation of the local elastic properties starts from the measure of particle-based elastic ones [49], which we obtain by studying how a per-particle defined stress tensor varies under imposed external deformations [41]. We provide details in Appendix A. The distribution of the single-particle shear modulus $\mu_i = \mu_{xy,i}$ [49] has a central Gaussian peak and long, asymmetric tails, as illustrated in Fig. 3a, where the moduli are scaled by their averaged value $\mu_0$ (equal to the macroscopic shear modulus). We previously observed analogous distributions in three dimensions [41]. Fig. 3a clarifies that the scaled variance $\gamma_1 = \sigma_{\mu_i}^2/\mu_0^2$ of the single-particle shear modulus decreases as $f$ increases.

The spatial heterogeneities of the local elastic constant are commonly assessed via the investigation of the disorder parameter $\gamma$ introduced in Schirmacher's fluctuating elasticity [3,

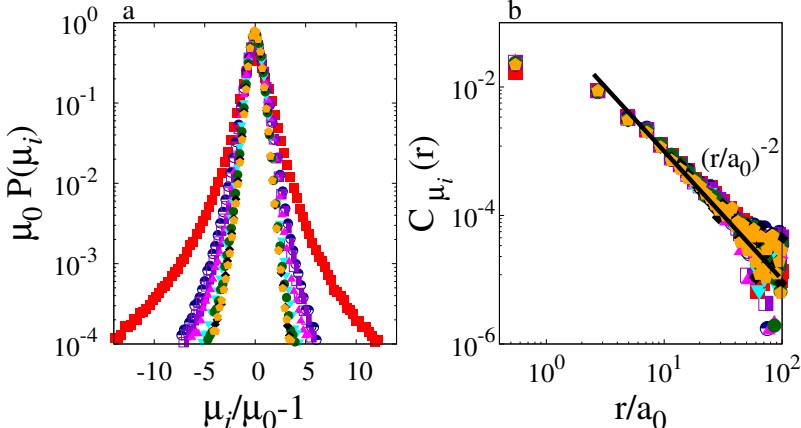

Figure 3: (a) Scaled distribution of the single-particle shear modulus. (b) Correlation function of the single-particle shear modulus defined in Eq. 3. Symbols represent different $f$ values as in Fig. 2d.

50]. In this theory, the elastic constants are assumed to be short range correlated. Therefore, according to central limit theory, the fluctuations $\sigma_w$ of the shear modulus coarse-grained over a large length scale $w$ scale asymptotically as the inverse of the number of particles in that region, $N_w \propto w^d$ in $d$ spatial dimensions. Therefore, the normalised fluctuations of the shear modulus asymptotically scale as

$$\frac{\sigma_w^2}{\mu_0^2} \propto \frac{\gamma}{N_w}, \tag{2}$$

where $\gamma$ is proportional to the fluctuations of the single-particle modulus and the correlation volume.

The assumption of spatially uncorrelated elastic constants does hold for amorphous solids. Indeed, stability induces long-range shear correlations ($\propto r^{-d}$ in $d$ spatial dimensions) with quadrupolar spatial symmetry in the shear stress [51] and the shear modulus [49, 52]. These correlations have been numerically observed in model systems [49, 51, 52] via the study of a radial correlation function that occurs for the quadrupolar symmetry [49, 52],

$$C(r) = \frac{\langle \mu(0)\mu(r) \rangle - \langle \mu \rangle^2}{\langle \mu^2 \rangle - \langle \mu \rangle^2} \cos(4\theta), \tag{3}$$

where $\theta$ is the angle between $\mathbf{r}$ and the $x$-axis (as $\mu = \mu_{xy}$). Our model amorphous networks reproduce these correlations, as illustrated in Fig. 3b. The figure demonstrates that curves corresponding to different $f$ values collapse onto each other. This observation indicates that the bond-swapping does not alter the elastic constant's spatial correlations, as conversely the asymptotic $r^{-2}$ decay would have set in after an $f$-dependent length.

While the local shear modulus has long-range correlations, the anisotropy of these correlations ensures that $\langle \mu(0)\mu(r) \rangle = 0$ at all $r$. Equivalently, the radial correlation function of the local shear modulus appears $\delta$-correlated if the quadrupolar symmetry is not taken into account. Because of this, the fluctuations of the shear modulus of $N_w$ particles enclosed in a compact volume are insensitive to anisotropic correlations and scale as if there were no correlations, Eq. 2. Indeed, many previous works verified Eq. 2, e.g. [4, 23, 41, 44, 53, 54].

The coarse-grained normalised fluctuations of the single-particle shear modulus of our model system also satisfy Eq. 2, as we demonstrate in Fig. 4a. The asymptotic value of the scaled fluctuations define the disorder parameter $\gamma$, we find to decrease as the degree of swapping $f$ increases, as in Fig. 4b. The disorder parameter depend on the fluctuations of the

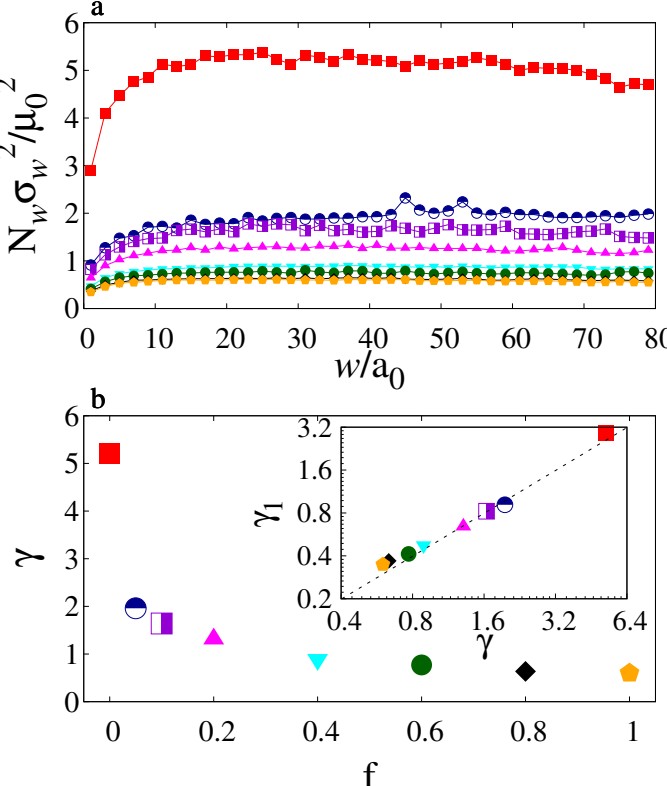

Figure 4: Panel a illustrates the dependence of fluctuations of the local shear modulus on the coarse-graining length scale $w$. Panel (b) shows the asymptotic value of Schirmacher's disorder parameter $\gamma$ as a function of swapping fraction $f$. The disorder parameter is proportional to the normalised fluctuations of the single-particle shear modulus. The inset illustrates that $\gamma$ is to a good approximation proportional to the normalised fluctuations of the single-particle shear modulus $\gamma_1$.

single-particle shear modulus and on its spatial correlations. Since Fig. 3b proves that spatial correlation are $f$-independent, we expect $\gamma$ to be proportional to the scaled fluctuations of the single-particle shear modulus, $\gamma \propto \gamma_1$, as we observe in the inset of Fig. 4b.

These investigations demonstrate that the local elastic properties of our model systems have the same correlations observed in amorphous materials, provided the elastic correlation length stays constant. The dependence of the disorder parameter $\gamma$ on $f$ indicates that swapping leads to the networks resembling those of glasses with increased stability.

## 5 Vibrational spectra

We now show that the bond-swapping algorithm leads to elastic networks whose vibrational properties exhibit a Boson peak and QLMs and investigate how these vibrational anomalies relate to the disorder parameter $\gamma$.

### 5.1 Boson peak

We determine the vibrational density of states of large $N = 160000$ systems by Fourier transforming the velocity auto-correlation function. Fig. 5(a) illustrates the vDOS for different $f$ values, upon scaling the frequency by $\omega_0 = c_s/a_0$, with $c_s = \sqrt{\mu_0/\rho}$ the shear-wave speed

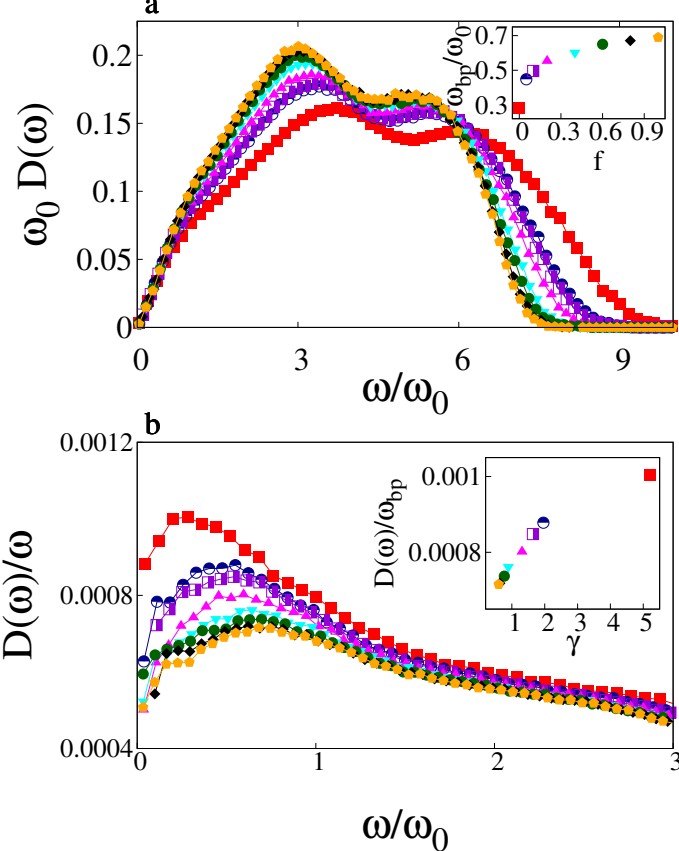

Figure 5: Panel (a) illustrates the density of states of system with $N = 160000$ as a function of $\omega/\omega_0$ for different swapping fractions $f$. The Boson peak frequency shifts towards higher frequency on increasing $f$. Panel (b) shows the reduced $D(\omega)$, as a function of $\omega/\omega_0$. The Boson peak strength ($D(\omega_{bp})/omega_{bp}$ increases with the disorder parameter, as in the inset.

and $a_0 = \rho^{-1/2}$ the interparticle spacing. The reduced vDOS $D(\omega)/\omega$ exhibits a boson peak at characteristic frequency which increases with $f$. Fig. 5(b) illustrates the reduced density of states, to highlight that the boson peak strength decreases as $f$ increases, or equivalently, the disordered parameter decreases, as illustrated in the inset. These results further confirm that swapping leads to networks resembling those of stable glasses, which have a reduced Boson peak anomaly [3, 7, 17, 41, 50].

## 5.2 Quasi-localized modes

We investigate the vibrational spectrum' low-frequency end via the direct diagonalization of the Hessian matrix. We focus on small $N = 1024$ systems to shift the lowest phonon frequency ($\omega_{min} \propto c_s/L$) upwards, exposing the QLMs, and perform averages over 50000 realizations for each $f$ value. For all swapping probabilities, $f$, QLMs are distributed in frequency as $A_4\omega^4$, as illustrated in Fig. 6a. The amplitude $A_4$ decreases on increasing $f$, as in the inset, again suggesting that networks with larger $f$ mimic those of glasses with increased stability.

Our swapping procedure leads to a small change in pre-stress $e$, which could be considered responsible [44] for the observed change in $A_4$. To assess this possibility, for some values of $f > 0$ we have prepared additional networks by slightly changing the density, and re-minimizing the energy, after the swapping procedure. We considered density changes that induce a net zero-change in the pre-stress, as shown with open symbols in Fig. 2d. Fig. 6

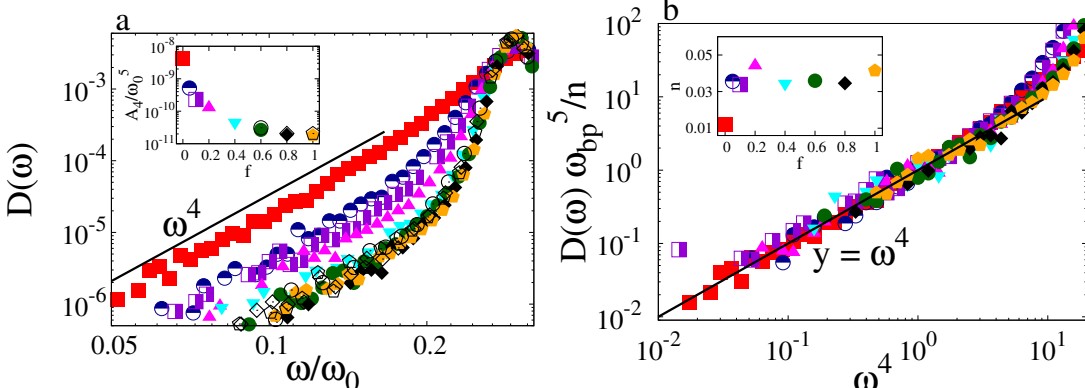

Figure 6: (a) $D_{\mathrm{loc}} = A_4 \omega^4$ scaling of low-frequency density of states. Different closed symbols indicate different swapping fraction $f$. The inset illustrates the $f$ dependence of the prefactor $A_4$. Open symbols for $f \geq 0.6$ represent compressed systems of constant prestress in both the main panel and the inset. (b) A plot of $D_{\mathrm{loc}} \omega_{\mathrm{bp}}^5 / n$ vs $\omega$, with $n$ a number density that weakly depends on $f$ as in the inset, leads to a data collapse. For each $f$, data are averaged over $50{,}000$ configurations of $N = 1024$ particles.

demonstrates that the low-density end of the vibrational density of states of these additional networks match those of the original ones, clarifying that small changes in the prestress do not affect the vibrational properties.

The amplitude $A_4$ has the units of a density of modes over a frequency to the power 5. If $\omega_{\mathrm{bp}}$ is the QLMs' characteristic frequency, then $A_4^{-1} = \omega_{\mathrm{bp}}^5 / n$ with $n$ the number density. In Fig. 6, we find that if $D(\omega)\omega_{\mathrm{bp}}^5 / n$ is plotted versus $\omega^4$ data for different $f$ collapse on a master curve, $A_4 \omega_{\mathrm{bp}}^5$ having a weak $f$ dependence, particularly for $f > 0$, as in the inset. This result establishes a close correspondence between Boson peak and QLMs. The estimated values of $n$ are essentially constant. A similar result holds in three-dimensional glasses [17, 41]. In that case, however, the density of modes $n$ resulted smaller by a factor of ten. However, Ref. [16] found more stable glasses to have a smaller $n$.

We estimate the typical size of a mode as $Np$, with $p$ the mode-participation ratio. Fig. 7 shows a crossover from extended modes, where $Np$ is of the order of $N = 1024$, to more localised ones as the frequency decreases. Interestingly, the localised modes involve around $Np \simeq 100$ particles, regardless of the swapping fraction $f$. This result is consistent with previous ones suggesting that the size of the localised modes relates to the correlation length of the elastic properties [41], as we do not see this length changing with $f$ (see Fig. 3b). Besides, the large asymptotic $Np$ value indicates that the low-frequency modes might not be truly localised. Indeed, previous results have shown that the lowest frequency modes have an $N$ independent $Np$ value in three and four spatial dimensions [55], qualifying them as localised. Conversely, $Np$ grows with $N$ in two-spatial dimensions, indicating that two-dimensional low-frequency modes are not truly localised.

# 6  Phonon attenuation

We now discuss how swapping influences phonon attenuation. To evaluate the phonon attenuation rate, $\Gamma$, we excite [56, 57] a transverse acoustic wave by giving each particle a velocity $\mathbf{v}_i^0 = \mathbf{A}_T \cos(\boldsymbol{\kappa} \mathbf{r}_i^0)$, where $\mathbf{A}_T \boldsymbol{\kappa} = \mathbf{0}$, considering $\boldsymbol{\kappa}$ in which one among $\kappa_x$ and $\kappa_y$ is zero, and

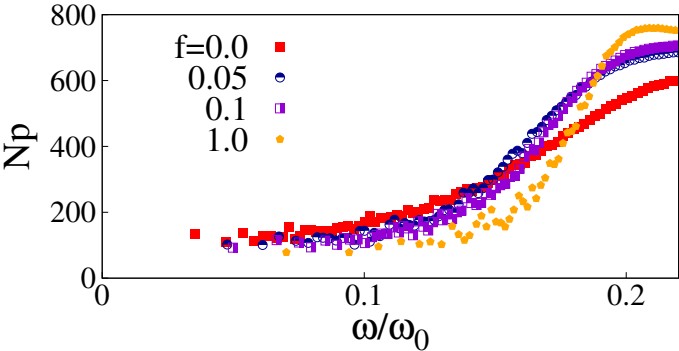

Figure 7: The volume associated with the localized modes ratio of modes in $N = 1024$ as a function of $\omega/\omega_0$. Data are averaged over 50000 realizations. For clarity, we show results for selected $f$ values.

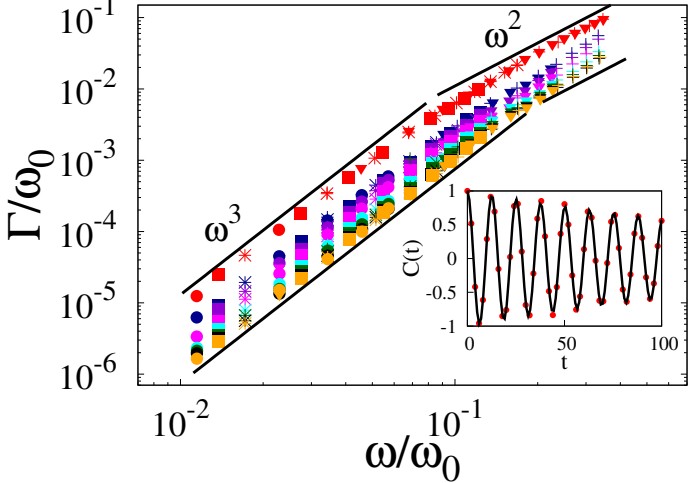

Figure 8: Dependence of the phonons' attenuation rate on the frequency normalized by $\omega_0$. Colors correspond to different swapping fractions $f$, as in Fig. 6b. Different symbols correspond to different system sizes, $N$=40000 (+), 90000 ($\triangledown$), 160000 ($\star$), 250000 ($\square$) and 360000 ($\circ$). The inset shows the velocity auto-correlation function $C(t)$ of transverse phonons of wavevector $k$ excited at $t = 0$, with superimposed damped cosine wave fit, for $N = 160000$, $f = 0.0$, and $\kappa^2 = 9(2\pi/L)^2$.

evaluate the velocity auto-correlation function:

$$C(t) = \frac{\sum_{i=1}^{N} \mathbf{v}_i(0).\mathbf{v}_i(t)}{\sum_{i=1}^{N} \mathbf{v}_i(0).\mathbf{v}_i(0)} . \tag{4}$$

We remind we work by definition in the linear response regime as we are considering the response of a system of masses and springs. For each $\kappa = |\boldsymbol{\kappa}|$, we average this correlation function over 30 phonons from independent samples for $N \leq 360000$. Finally, we extract attenuation rate $\Gamma$ and frequency $\omega$ as a function of wave-vector $\kappa$ by fitting the velocity auto-correlation function to a damped oscillation, $\cos(\omega t)e^{-\Gamma t/2}$. As an example of this procedure, we show in the inset of Fig 8 the velocity autocorrelation function for $\kappa = \frac{2\pi}{L}(3,0,0)$ in a $N = 160000$ system and its damped exponential fit.

Fig 8 illustrates the dependence of the attenuation parameter on $\omega/\omega_0$. At all $f$ values, we observe the crossover from strong Rayleigh scattering $\Gamma \sim \omega^3$ to disorder-broadening regime $\Gamma \sim \omega^2$ with increasing frequency $\omega$ as found in glasses. At fixed $\omega/\omega_0$, the attenuation

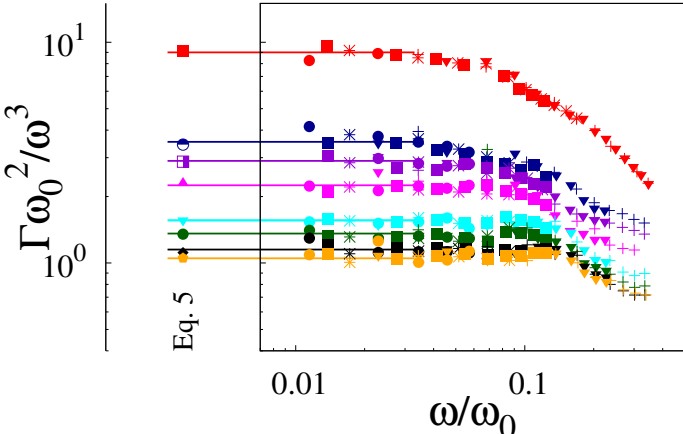

Figure 9: Scaled attenuation rate as a function of the frequency. The asymptotic value in the $\omega \to 0$ limit is compared to the predictions of classical FET. This prediction corresponds to Eq. 5 with $\xi$ constant, as FET assumes the local elastic properties to be $\delta$-correlated. Symbols identify the system size as in Fig. 8.

parameter $\Gamma$ reduces with increasing $f$. We remark that these results do not suffer from size effects, as we explicitly show combining data for different N values.

Rayleigh's original model [24] explains the $\omega^{d+1}$ scaling of the attenuation rate by describing the elastic medium as an elastic continuum punctuated by isolated defects. In this model, the attenuation rate depends on the number density of defects $n$, their size $\xi$, and the deviation of their shear elastic properties from the background. In terms of the disorder parameter $\gamma$, Rayleigh's prediction results [41]

$$\Gamma \frac{\xi}{c_s} \propto \gamma \left( \frac{\omega \xi}{c_s} \right)^{d+1} . \tag{5}$$

The same prediction is recovered by a version of fluctuating elasticity theory that considers the elastic properties to be correlated over a length scale $\xi$ [41, 50]. In the present system two-dimensional system, at variance with the three-dimensional case, the typical size of the localised modes is constant. This suggests that $\xi$ is constant so that Rayleigh's prediction simplifies to $\Gamma \frac{a_0}{c_s} \propto \gamma \left( \frac{\omega a_0}{c_s} \right)^4$. This prediction is obtained by fluctuating elasticity theory at constant elastic correlation length [3]. Eq. 5 with $\xi$ constant has also been recovered by studying the disordered induced broadening of the width of phonon bands [58].

Fig. 9 illustrates the dependence of $\Gamma \omega_0^2 / \omega^{d+1}$ on $\omega/\omega_0$, and demonstrates that the low-frequency attenuation rate is well described by the theoretical prediction of Eq. 5 with $\xi$ constant. This result confirms previous investigation of sound attenuation in two-dimensional systems [44]. In three dimensions, Eq. 5 also holds, but the correlation length $\xi$, identified with the size of the soft modes, decreases as the glass becomes more stable [41].

Eq. 5 predicts the dependence of the attenuation rate on the disorder parameter and the elastic correlation length up to a constant factor. Estimating this factor suggests that the fluctuating elasticity framework quantitatively underestimates the damping coefficient [59]. Possibly, this occurs as the theory uses linear elasticity to account for the scattering by sources of length scale $\xi$, without considering that elasticity theory only holds on larger length scales [16]. In addition, fluctuating elasticity theory assumes the local elastic constant to be short-range correlated, while conversely, they are long-range correlated, as we illustrated in Fig. 3b.

## 7 Conclusions

We have introduced an algorithm that modulates the elastic disorder of a mass-spring network without affecting its connectivity by swapping the properties of randomly selected bonds. We have generated a series of disordered networks and investigated their elastic properties using this algorithm and an initial network derived from a two-dimensional model system. Increasing the fraction of swapped bonds suppresses the fluctuation of the shear modulus, reduces the Boson peak's amplitude and the Boson peak's frequency over the natural frequency $\omega_0$, and increases the typical frequency of the low-frequency modes. As such, bond-swapping leads to networks whose vibrational properties increasingly resemble those of stable glasses. The observed changes in vibrational properties do not relate to variation in the connectivity occurring close to the jamming point [60], or analogous, to the emergence of many weak contacts leading to a small effective connectivity [61]. Similarly, they do not originate from changes in the prestress.

The generated networks reproduce the sound wave attenuation rate's crossover from a low-frequency Rayleigh scattering to a disordered broadening regime characterizing amorphous solids in the harmonic approximation. The attenuation rate in the Rayleigh scattering regime scales with the material properties as predicted by fluctuating elasticity theory, which works under the constant shear modulus correlation length assumption.

Our investigation of the bond-swapping algorithm to a two-dimensional model network reproduces vibrational anomalies and sound attenuation observed in two-dimensional amorphous solids. Soft modes are dimensionality dependent as in two-dimensions they are not truly localised [14, 62], a feature possibly connected to the dimensionality dependence of the glasses' relaxation dynamics [63–66]. It would be interesting to assess if the bond-swapping algorithm reproduces three-dimensional systems' vibrational properties. More generally, the influence of dimensionality on sound attenuation requires further investigations.

## Acknowledgments

We thank E. Lerner and E. Bouchbinder for comments on a earlier version of this manuscript.

**Funding information** We acknowledge support from the Singapore Ministry of Education through the Academic Research Fund Tier 1 (RG56/21) and Tier 2 (MOE-T2EP50221-0016) and are grateful to the National Supercomputing Centre (NSCC) of Singapore for providing the computational resources.

## A  Local elasticity

Within two-dimensional linear elasticity the macroscopic stress and the macroscopic strain are related by $\sigma_\alpha = c_{\alpha\beta}\epsilon_\beta$, with $\alpha, \beta \in \{xx, yy, xy\}$ and $c_{\alpha\beta}$ the stiffness tensor. Here, we define the local stiffness matrix $c_{\alpha\beta}^{\mathrm{cg}} = \frac{d\sigma_\alpha^{\mathrm{cg}}}{d\epsilon_\beta}$ as the ratio between a locally defined stress and the macroscopic strain [41, 52]. We define the coarse grained stress as $\sigma_\alpha^{\mathrm{cg}}(w) = \langle \sigma_\alpha^{(i)} \rangle$, where the average is over all particles $i$ in the coarse-graining volume, and $\sigma_\alpha^{(i)}$ is a per-particle stress [49].

We note that other approaches could be used to define local elastic properties [67, 68], e.g., by introducing locally defined stresses and strains. These diverse definitions converge for large coarse-gaining lengths but differ at finite $w$. The definition we have adopted here

recover self-averaging. In addition, with this definition the statistics of the elastic properties coarse grained over a sub-region containing $N_0$ particles of a $N \gg N_0$ system match those of a $N_0$ particle system [41].

Practically, we evaluate $c_{\alpha\beta}$ by monitoring the change in the stresses of the particles in response to small deformation followed by energy minimization, to capture the non-affine contribution to the elasticity, making sure we work in the linear response regime. We coarse grain the single-particle elastic properties over square regions of side length $w$.

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
