# Peer review of "Quasi-localized vibrational modes, Boson peak and sound attenuation in model mass-spring networks"

_SciPost Physics, doi:SciPost Phys. 15, 069 (2023)_

## Round 2 · Referee Report · Anonymous (Referee 1) · 2023-4-10

Report

In the revised manuscript, the authors have done substantial additions in the text along with figure modifications, in response to the referee reports. The article reports new analysis of shear modulus spatial fluctuations/correlations and a radical change of conclusion, namely that the swapping algorithm does not affect the glassy length scale.

Although I feel that the article should be published in SciPost, there are still some points that need to be addressed. I feel some details have been overlooked in the presentation of the setup and the analysis. Please find below some remarks.

Requested changes

Shear modulus fluctuations/correlations: If I understand correctly, the method to extract the spatial map of the shear modulus in both V1 and V2 is the same, i.e. an affine deformation followed by an energy minimization with a stress field coarse-grained on a scale w.

  • Authors present the scaled \sigma_w in Fig.4(a) but the curves (especially for f=0) have now changed between V1 and V2. Why so?

  • Authors now report the P(\mu_i), where \mu_i is the shear modulus at the particle level. On which scale w is it calculated? Should P(\mu_i) show more asymmetric tails between low and high shear moduli? Does P(\mu_i) exhibit the correct asymptotic scaling for small \mu_i?

  • Authors claim that the spatial correlation of \mu_i shows long range correlations, which to my knowledge has not been highlighted in PHYSICAL REVIEW E 80, 026112 (2009) and other related studies. I am confused about Fig.3(b) and the discussion around it. It starts by citing Ref.50 which only focuses on stress correlations, although the text reads “long-range correlations in the local shear modulus”, same for Ref.56. Could you please either change this statement or explain it a bit better.

  • The text reads “the anisotropy of these correlations ensures that \langle\mu(0)\mu(r)\rangle=0 at all r”, which quantity is then reported in Fig.3(b)?

  • Authors states “the fluctuations of the shear modulus of N_w particles enclosed in a compact volume are insensitive to anisotropic correlations and scale as if there were no correlations, Eq. 2” Should then the data in Fig.4(a) be flat at all scales w?

Glassy length scale: Previous works (including many contributions by the authors) in both 2D and 3D have also put forward the relation between \xi \sim 1 / \omega, where \omega can be viewed as the typical excitation frequency or Boson peak frequency. In the present study \omega_{BP} only changes by a factor 2-3, accordingly, one needs to look for a factor 2-3 decrease of the glassy length scale as a function of f. I genuinely do not believe that neither Fig.3(b) nor Fig.6 provide for the evidence that \xi is constant.

  • As mentioned above I would first clarify what is plotted in Fig.3(b), and then only report f=0 and f=1 for clarity. We cannot appreciate if \xi is constant when plotting all values of f in log-log scale.

  • Additionally, Fig.4(a) shows that the data for f=0 converges more slowly than f>0. One expects such a result, as the spatial extent of non-affinities scale with \xi. I find results in Fig.4(a) consistent with a decrease of \xi as f increases as shown in the V1.

  • The quality of the data of Fig.6 is not satisfying enough to draw any conclusion either. Inspecting closely the data, one actually finds a decrease of the modes participation ratio between f=0 and f=1. Again, we are looking for a factor 2-3, which on the y-log scale of Fig.6 cannot be ruled out.

  • I do not understand how the error bars are computed in Fig.6. There is a lot of fluctuation and some points have very large errors and some don’t. I do understand the computational difficulty and therefore I surely do not ask authors to improve the statistics, but I would at least only report average Np values for bins that contain at least 10 modes and evaluate the spread as a function of the frequency. One could also only report f=0 and f=1 for clarity if it helps.

  • I would also add a note of caution in the main text, because the participation ratio is ill defined in 2D due to the log correction, i.e. Np is size dependent.

  • validity: -
  • significance: -
  • originality: -
  • clarity: -
  • formatting: -
  • grammar: -

Author:  Massimo Pica Ciamarra  on 2023-04-14  [id 3586]

(in reply to Report 1 on 2023-04-10)
Category:
answer to question

We thank the reviewer for the additional comments. We have clarified the points of concern in the revised manuscript.
W provide a point-to-point response to the reviewer's comment in the attached file

Attachment:

reply_YfVKF5q.pdf

Author:  Massimo Pica Ciamarra  on 2023-04-14  [id 3584]

(in reply to Report 1 on 2023-04-10)

We thank the reviewer for the additional comments to which we reply in the attached file.

Attachment:

reply.pdf

---

## Round 2 · Referee Report · Anonymous (Referee 2) · 2023-4-11

Strengths

Same as in my first report.

Weaknesses

Same as in my first report.

Report

I have read through the authors' report. They reply properly to all of my questions and suggestions. The manuscript has improved a lot. The subject of this paper is solid-state properties of glasses, which is one of active and important topics in physics of amorphous materials. This paper can contribute to this subject, and thus I recommend publication in the current form.
  • validity: high
  • significance: good
  • originality: good
  • clarity: high
  • formatting: excellent
  • grammar: excellent

Author:  Massimo Pica Ciamarra  on 2023-04-14  [id 3585]

(in reply to Report 2 on 2023-04-11)

We are glad the reviewer is satisfied with our revised version and thank him for his help in improving our work.

---

## Round 2 · Author Response

Dear Editor,

We thank the reviewer for their valuable comments.
To clarify their concerns, we have performed additional studies, including a study of the spatial correlations of the local elastic properties and a detailed study of the influence of prestress on the vibrational properties.
The revised manuscript has been modified to detail these additional studies, which bring clarity to our findings.

Sincerely,
the authors

---

## Round 2 · List of Changes

List of changes - we added a figure reporting the spatial correlation function of the local elastic properties - we clarified that our algorithm does not modify the elastic correlation length - we discuss the dependence of the Boson peak strength on the main control parameter of our algorithm - we have performed additional simulations to compare the elastic properties of disordered mass-spring networks at fixed prestress. - we have added a few references as suggested by the referee and clarified the limitation of the FET picture in predicting sound attenuation quantitatively - we have changed the text throughout the paper to illustrate the results of the novel investigations.

---

## Round 3 · Author Response

Dear Editor,

We thank the reviewer for having considered our revised manuscript.
We are glad Ref 2 considers our manuscript suitable for publication and that Ref 1 considers our manuscript deserving of publication after clarifying a few points.

We have clarified these points in the revised manuscript and provided a detailed reply to the comments by Ref 1.

Sincerely,
Massimo

---

## Round 3 · List of Changes

• We made minor changes to improve clarity, as the reviewer's comments suggest we did not properly define some of the quantities we studied.

  • We revised Fig. 7. Following the reviewer's suggestion, we changed the procedure to bin the data and show results for selected values of f to improve clarity.

---

## Round 4 · Author Response

We are submitting V4 to solve a technical glitch. This version is identical to V3.

---

## Editorial Decision

published